

# Machine learning and natural language processing to assess the emotional impact of influencers' mental health content on Instagram

Noemi Merayo[1], Alba Ayuso-Lanchares[2] and Clara González-Sanguino[3]

[1] Signal Theory, Communications and Telematic Engineering Department, High School of Telecommunications Engineering, Universidad de Valladolid, Valladolid, Valladolid, Spain
[2] Department of Pedagogy, Faculty of Medicine, Universidad de Valladolid, Valladolid, Valladolid, Spain
[3] Department of Psychology, Education and Social Work Faculty, Universidad de Valladolid, Valladolid, Valladolid, Spain

Corresponding author
Noemi Merayo, noemer@tel.uva.es

## ABSTRACT

**Background:** This study aims to examine, through artificial intelligence, specifically machine learning, the emotional impact generated by disclosures about mental health on social media. In contrast to previous research, which primarily focused on identifying psychopathologies, our study investigates the emotional response to mental health-related content on Instagram, particularly content created by influencers/celebrities. This platform, especially favored by the youth, is the stage where these influencers exert significant social impact, and where their analysis holds strong relevance. Analyzing mental health with machine learning techniques on Instagram is unprecedented, as all existing research has primarily focused on Twitter.
**Methods:** This research involves creating a new *corpus* labelled with responses to mental health posts made by influencers/celebrities on Instagram, categorized by emotions such as love/admiration, anger/contempt/mockery, gratitude, identification/empathy, and sadness. The study is complemented by modelling a set of machine learning algorithms to efficiently detect the emotions arising when faced with these mental health disclosures on Instagram, using the previous *corpus*.
**Results:** Results have shown that machine learning algorithms can effectively detect such emotional responses. Traditional techniques, such as Random Forest, showed decent performance with low computational loads (around 50%), while deep learning and Bidirectional Encoder Representation from Transformers (BERT) algorithms achieved very good results. In particular, the BERT models reached accuracy levels between 86–90%, and the deep learning model achieved 72% accuracy. These results are satisfactory, considering that predicting emotions, especially in social networks, is challenging due to factors such as the subjectivity of emotion interpretation, the variability of emotions between individuals, and the interpretation of emotions in different cultures and communities.
**Discussion:** This cross-cutting research between mental health and artificial intelligence allows us to understand the emotional impact generated by mental health content on social networks, especially content generated by influential celebrities among young people. The application of machine learning allows us to understand the emotional reactions of society to messages related to mental health, which is

highly innovative and socially relevant given the importance of the phenomenon in societies. In fact, the proposed algorithms' high accuracy (86–90%) in social contexts like mental health, where detecting negative emotions is crucial, presents a promising research avenue. Achieving such levels of accuracy is highly valuable due to the significant implications of false positives or false negatives in this social context.

## INTRODUCTION

The growing popularity of social networks, along with the increasing relevance of mental health and its unstoppable presence on social media, are creating an environment of growing concern. On the one hand, social networks have become the most representative tool of the Internet, generating a continuous flow of information that reflects, in real time, the attitudes and trends of our society. As of October 2023, there were approximately 4.95 billion social media users globally, constituting roughly 61% of the world's population (*Kempt, 2023*), and people spends around 2½ h per day using social platforms. Although WhatsApp is globally the favourite social network among all users (16.6%), Instagram closely follows in second place (17.7%). In fact, Instagram shows significant adoption among young people, being the favourite among both female and male users aged 16–24 (24.4%), and among female (20%) and male (16.3%) users aged 25–34. Instagram has witnessed an increase in its share among males aged 16 to 24 in recent years. In the case of Spain, where this study took place, Instagram ranks as the second favorite social platform among the Millennial generation (74% of users), that is, those between the ages of 25 and 34 (*Statista, 2023*).

On the other hand, mental health problems currently affect a significant percentage of the population. According to the World Health Organization, one in four people will suffer from some kind of mental health problem or neurological disorders at some point in their lives. Regarding youth, one in seven individuals aged 10 to 19 suffers from a mental disorder. Moreover, depression, anxiety, and behavioral disorders are the main mental health disorders, and suicide is the fourth leading cause of death among those aged 15 to 29 (*World Health Organization, 2021*).

In recent years, there has been an unstoppable increase in the importance of mental health, especially in more developed societies (*Asper et al., 2022*). A progressively common phenomenon is the public self-disclosure of mental health issues by celebrities and influencers, particularly in virtual environments where the reach is challenging to control (*Francis, 2021*; *Gronholm & Thornicroft, 2022*; *World Health Organization, 2021*).

Despite this strong social presence, it is also clear that negative attitudes and prejudice towards mental health problems and disorders still exist. This negative attitude towards mental health refers to beliefs, prejudices, negative emotions and discrimination

surrounding individuals with mental disorders. It can manifest as social rejection, fear, lack of understanding and marginalisation of people (*Corrigan & Watson, 2002*). Although the use of social media to address mental health could contribute to understanding and combating misconceptions or associated prejudices, as shown by studies in which celebrities share their mental health problems and raise awareness (*Gronholm & Thornicroft, 2022*; *Jain, Pandey & Roy, 2017*; *Lee, 2019*; *Lee, Yuan & Wohn, 2021*), the sheer volume of opinions expressed on social media have created a complex phenomenon to assess. Consequently, it is unclear whether such posts truly foster acceptance or, conversely, perpetuate greater trivialization of mental health, as demonstrated by other studies (*Pavlova & Berkers, 2022*; *Robinson et al., 2019*).

In this social context, the use of artificial intelligence (AI) and machine learning (ML) can revolutionize the field of mental health, as it enables the quick and efficient automation, classification, and analysis of information. Through sentiment analysis, which merges natural language processing (NLP) and Computational Linguistics (*Mäntylä, Graziotin & Kuutila, 2016*), it becomes possible to detect the emotional tone of comments made in response to a post. This allows real-time trend tracking, understanding the emotional response of society to a specific topic and predicting the future behavior of users. Its integration into mental health topics on social media holds significant potential and impact, in addition to being an innovative and pioneering interaction. Thus, sentiment analysis can be performed using different techniques.

One of the most widely used approaches in sentiment analysis involves the use of lexicons. A sentiment lexicon is a linguistic database that assigns labels of emotional polarity (positive, negative, or neutral) or specific emotions (love, hate, sadness, *etc.*) to words and expressions, thereby facilitating sentiment analysis in texts by identifying and quantifying the emotions conveyed through language. Although lexicons are useful for sentiment analysis, they present important disadvantages, such as: not fully capturing the context in which words are used, semantic ambiguity, as some words can have multiple meanings; difficulty with idiomatic expressions and colloquial phrases due to their figurative or non-literal meaning; problems with understanding irony and sarcasm, which can reverse the meaning of words; difficulty with neutral or polysemous words that are challenging to categorize; lack of consideration for complex grammatical structures, which can sometimes significantly affect the overall sentiment; and finally, limitations in detecting negation (*Taboada et al., 2011*). Besides, within the Spanish language setting, the presence of these dictionaries is restricted (*Redondo et al., 2007*). Despite these disadvantages, lexicons are valuable tools when combined with other approaches, such as machine learning, to address the complexity of sentiment analysis in natural language.

On the other hand, supervised learning (ML) with labelled corpora for sentiment analysis involves training artificial intelligence models using datasets previously labelled with the emotions associated with each text. This allows models to learn patterns and features accurately associated with different emotional states (*Li & Yi, 2022*; *Plaza-Del-Arco et al., 2021*; *Shah et al., 2023*). The advantages include the ability to enhance the accuracy and efficiency of sentiment analysis by providing clear and specific training examples, enabling the model to generalize and classify emotions more accurately in

unlabelled texts. Indeed, some research studies (*Srivastava, Bharti & Verma, 2022*) have demonstrated the effectiveness of supervised approaches over lexicon-based approaches. For all these advantages, our research has been oriented towards machine learning techniques. Moreover, sentiment analysis on social media poses significant challenges for artificial intelligence algorithms (*Martínez-Cámara et al., 2014*), including the brevity and lack of context in messages, as well as the use of informal grammar, idiomatic expressions, and abbreviations. These characteristics make the application of effective techniques more complex, as language on social media significantly differs from conventional usage.

In this way, machine learning and NLP methods can provide diverse perspectives in mental health research since they have the capability to extract valuable insights from data that may otherwise be unexplored or inaccessible to mental health professionals (*Le Glaz et al., 2021*). Thus, text analytics have gained significant momentum in the health sector, to automate the process of extracting information from unstructured textual data (*Elbattah et al., 2021*). The systematic review by *Zhang et al. (2022)* reveals that there is a strong trend in research integrating NLP for mental illness detection, where deep learning methods perform better than traditional methods. In this way, the analysis carried out by *Malgaroli et al. (2023)* stated a rapid growth in natural language processing studies in mental health since 2019.

However, there are few studies that incorporate machine learning and NLP in the analysis of mental health on social networks, especially in the Spanish context and on one of the platforms most popular among young people, such as Instagram. In fact, the use of artificial intelligence and natural language processing in the field of mental health in social networks has focused over time on the assessment and detection of mental disorders and psychopathology (*Iyortsuun et al., 2023*; *Khan & Ali, 2024*). This application has been used both in clinical samples of already diagnosed individuals, as well as in social network users from the general population (*Le Glaz et al., 2021*).

Furthermore, the main social network on which research has been conducted is Twitter (now X) (*Di Cara et al., 2023*), even though it is not the most widely used social network at present. Instagram is emerging as one of the most popular social networks worldwide, especially among young people. Although Instagram was initially conceived solely for image posting, it now serves as a means of communication where, while the image is important, the accompanying text and messages reacting to it reflect opinions, preferences, tastes, and, in general, various topics. Besides, there are no studies related to the analysis of sentiment and its impact when discussing this type of symptomatology in social networks, especially when conducted by authors who can exert a strong social impact on millions of users worldwide, such as celebrities or influencers.

In this research context, which focuses on analyzing emotional responses (emotions) on Instagram within the Spanish context, particularly in relation to the disclosure of mental health issues, it emerges as an exceptional environment with significant impact. In fact, the analysis and detection of emotions is particularly difficult due to different challenges, such as individual subjectivity in the interpretation of emotions, the diversity of emotions in different cultures and communities, or the way different people interpret their emotions. This setting provides an opportunity to understand public opinion and emotional

responses triggered by these disclosures, especially when conveyed by people (celebrities, influencers) who reach millions of people. Consequently, our research includes two main contributions:

(1) To develop an innovative *corpus* labelled with emotions extracted from comments on Instagram posts related to celebrity disclosures about mental health. This *corpus*, together with the decalogue for categorising emotions and the dataset, is accessible in a GitHub repository (*GitHub, 2024*) for researchers.

(2) To model machine learning algorithms (*GitHub, 2024*) to identify emotions expressed on Instagram in the context of mental health disclosures, particularly those related to anxiety or depression.

This research marks a milestone by applying, machine learning to mental health issues on Instagram, in contrast to the majority of analyses focused on Twitter. Furthermore, our proposal focuses, in a pioneering way, on analysing emotional response in social networks, in contrast to existing research that focuses mainly on identifying types of psychopathology (*Ahmed et al., 2022*; *Birnbaum et al., 2017*; *Fodeh et al., 2019*; *Guntuku et al., 2017*; *Joshi & Kanoongo, 2022*; *Lejeune et al., 2022*). Finally, our proposal has great potential to address the phenomenon of responses to the revelations of influential people, especially among young people, about mental health problems and whose impact reaches millions of people in the virtual world. Incorporating these AI technologies into mental health studies could advance our understanding of specific processes, enabling the implementation of more effective strategies. This, in turn, would promote broader awareness and have a positive impact on both society at large and specific groups, such as mental health professionals and institutions. Additionally, it would encourage responsible use of social media when addressing mental health topics.

This article is organized as follows. "State of the Art" details a background. "Dataset of Mental Health in Instagram" describes the dataset of mental health in Instagram. "Machine Learning Classification Models" shows the selected machine learning classification models to predict emotions. Besides, "Results and Analysis" describes the results and analysis of the classification models and "Discussion" the discussion of results. Finally, "Conclusions" summarises the main conclusions of the research.

## STATE OF THE ART

In recent years we are experiencing an increasingly frequent phenomenon, where celebrities and influencers make public their mental health problems, especially on social media. Specifically, the authors in *Gronholm & Thornicroft (2022)* analyse the impact generated by celebrities when they publish information about their experience of mental health illnesses, with the aim of determining whether and how these actions can reduce stigma. In this way, *Jain, Pandey & Roy (2017)* analyse the impact of a public announcement by one of Bollywood's most popular celebrities about her depression problems, and how it may influence awareness and perception of mental health in India, due to the strong popularity of the world of Bollywood.

Regarding the presence of this phenomenon in social networks, the authors of *Lee, Yuan & Wohn (2021)* analyzed how disclosures about streamers' depression issues could impact viewers' perceptions of these problems. They concluded that there is a strong association between disclosures about streamers' health and public awareness of depression. Moreover, the study presented at *Francis (2021)* analyzed conversations on Twitter following hip-hop artist Scott "Kid Cudi" Mescudi's disclosure of depression. This research has the potential to visualize the mental health experiences of black men and offer tools that could be utilized in mental health interventions for this demographic.

However, none of these research studies on the disclosure of mental health problems of celebrities integrate artificial intelligence techniques. Furthermore, in addition to this, the literature on mental health and artificial intelligence in social networks primarily focuses on the detection of disorders such as depression or anxiety (*Hasib et al., 2023*; *Khan & Alqahtani, 2024*; *Liaw & Chua, 2022*; *Pande et al., 2024*; *Vasha et al., 2023*, *Wongkoblap, Vadillo & Curcin, 2021*), schizophrenia (*Tyagi, Singh & Gore, 2023*), suicide (*Malhotra & Jindal, 2024*; *Rabani et al., 2023*; *Ramírez-Cifuentes et al., 2020*), or even to carry out a classification of different diagnoses (*Nova, 2023*; *Chung & Teo, 2023*), and most of this research has been centered around Twitter.

Thus, in the context of depression, several studies have developed deep learning and machine learning models to identify users with depression on Twitter. For instance, *Wongkoblap, Vadillo & Curcin (2021)* developed a deep learning model, while *Liaw & Chua (2022)* employed machine learning models that incorporate specific user characteristics influencing detection. *Khan & Alqahtani (2024)* applied machine learning models to detect signs of depression on Twitter using different metrics. *Hasib et al. (2023)* also utilized machine learning and deep learning techniques to enhance detection. Similarly, *Pande et al. (2024)* developed a system that utilizes machine learning, natural language processing, and user profiling to detect individuals at risk of depression based on their social media posts. Furthermore, *Vasha et al. (2023)* conducted research to identify depressed individuals through their comments and posts on social networks such as Facebook and YouTube, using support vector machines (SVM).

Regarding detection of different mental disorders, *Nova (2023)* demonstrated the effectiveness of machine learning models in the classification of mental disorders, in particular, bipolar, depression, anxiety and borderline personality disorder (BPD), using text data from social networks. *Chung & Teo (2023)* evaluates several machine learning algorithms for classifying and predicting mental health problems from surveys.

Additionally, some research has been devoted to identifying suicidal behaviour on social networks. A particular study, outlined in *Ramírez-Cifuentes et al. (2020)*, assessed the suicide risk of Spanish-speaking users on social networks using machine learning, based on data extracted from social networks (comments, posting patterns, relationships with other users and images posted). Besides, research by *Rabani et al. (2023)* proposed machine learning classifiers (SVM, Extreme Gradient Boosting, Random Forest) to identify levels of suicide risk in Twitter. Similarly, *Malhotra & Jindal (2024)* applied pre-trained Large Language Models (LLMs) to detect suicidal behaviour from social network users. Finally, *Tyagi, Singh & Gore (2023)* analyse machine learning and deep learning techniques for the

diagnosis of schizophrenia by gathering knowledge from different types of schizophrenia modalities.

Despite the rapid development of research focused on the detection of mental disorders using artificial intelligence, studies focused on the social response to them and the impact at the emotional level are very scarce. For instance, *Alvarez-Mon et al. (2021)* proposed exploring Twitter posts related to antipsychotic medications to better understand responses and identify common areas of clinical interest. *Jilka et al. (2022)* aims to reliably identify stigmatizing tweets and comprehend public stigma of schizophrenia on Twitter by employing machine learning algorithms. *Oscar et al. (2017)*, applied machine learning techniques to model the stigmatization expressed on Twitter in relation to Alzheimer's disease. *Bograd, Chen & Kavuluru (2022)* explored the feelings towards obesity in Twitter, and *Xue et al. (2020)* analysed the response to COVID-19. Finally, other transversal studies focus on the development of AI-based chatbots and ChatGPT (*Van der Schyff et al., 2023*, *Singh, 2023*) to provide support for mental health problems.

From this analysis of the state of the art, it can be inferred that there are a limited number of studies that incorporate AI and machine learning in the exploration of sentiment analysis and social response to mental health problems on social networks, particularly in the Spanish context and on platforms widely used by young people, such as Instagram.

# DATASET OF MENTAL HEALTH IN INSTAGRAM

## Selection of Instagram posts

The search for Instagram posts to be selected for our research study followed these premises:

a) Posts had to be made directly by the author and related to the disclosure of mental health issues.

b) Posts had to be made by Spanish celebrities and influencers and written in Spanish. The selection of these celebrities was based on their ability to have a significant impact, so it was stipulated that they should have more than 100,000 followers on their Instagram profiles.

To conduct this search, profiles of celebrities and influencers on Instagram were analyzed, and relevant news on these topics was reviewed in the press, television, and social media. The search ran from September 2022 to December 2022.

It is noteworthy that, following the search and identification process of posts, the majority of them were made by women. Therefore, the decision was made to exclude the male gender to prevent potential gender biases that could impact AI algorithms. However, in the future, it is planned to integrate the male perspective. Furthermore, another important challenge is the effectiveness of public participation strategies, especially in social networks, due to their inherent biases and the demographic diversity of users. Some groups may not actively participate in social networks, which could lead to biases in the collected data. This is an inherent risk of the research study, as it is primarily aimed at addressing mental health in virtual spaces.

Following these criteria, 20 posts made by women were identified, ultimately selecting the top 10 posts with the most comments. These posts and their associated comments were retrieved using the software tool (*Google, 2024*). Furthermore, all selected posts exhibited the following characteristics:

a) The format was consistent, featuring either a photo (five posts) or a video (five posts) accompanied by relevant text.
b) The content was either related to the manifestation of symptoms of a mental health problem or emphasized the importance of specialized care in such cases.
c) The most prevalent mental health issues addressed were anxiety and depression.

On the other hand, the ethical aspects of artificial intelligence in the social sciences is a topic of growing interest (*Mangino, Smith & Finch, 2021*), and the detection of disorders based on comments in social networks poses certain ethical problems in relation to privacy and good practices (*Fleming, 2021*). However, the objective of the present study is to find out what the reaction to the revelation of a disorder by celebrities is like. In other words, we are trying to find out what the social response to mental health in networks is like, not to make a diagnosis or detect psychopathology. This may allow us to promote in the future a more appropriate use of social networks in relation to mental health, as well as to raise awareness in society about the importance of mental health and its appropriate treatment.

In addition, this study has been carried out taking into account the recommendations of various authors for working with artificial intelligence in social sciences, such as making the results interpretable or being transparent in the development of the algorithms and database (*Mangino, Smith & Finch, 2021*). Additionally, regarding ethical considerations and data privacy, the study has been approved by the ethics and deontology committee of the University of Valladolid (PI 23-3365). Thus, only public posts and comments were accessed and never private accounts or conversations and, to protect privacy, we anonymised all Instagram comments, filtering out all @mentions and URLs. Table S1 presents a description of the selected Instagram posts, including the number of responses associated with each post, the names of the celebrities/influencers, and their respective number of followers. A total of 21,151 comments were gathered, and the process of categorizing, processing, and incorporating them into a mental health *corpus* will be describe in next sections.

## Description of dataset labelling in emotions

Following previous literature on manual text labelling (*Alvarez-Mon et al., 2021*; *Bograd, Chen & Kavuluru, 2022*; *Budenz et al., 2020*; *Delanys et al., 2022*) we developed guidelines on the different categories of labelling, emotions in our specific case (*GitHub, 2024*). Classification began based on the model of primary emotions: fear, anger, sadness, disgust, happiness and surprise (*Ekman, 2004*). However, this was insufficient to classify the positive emotions reflected in many comments. This is consistent with previous studies, which point out how basic emotions can be oversimplified representations of complex phenomena and lead to lower inter-annotator agreement when categorising in the context

of emotive language analyses (*Williams et al., 2019*). Thus, the categories of positive emotions were expanded based on the model of *Plutchik (2001)* and the approaches of *Fredrickson (2013)*. Finally, the categories proposed were love/admiration, gratitude, comprehension/empathy/identification, sadness, anger. A deep explanation of each emotions reads as follows:

- Love and admiration: Emotions present in *Fredrickson (2013)* and *Plutchik (2001)* models where admiration, approval and love are closely related. They are usually messages with positive content with praise, affection and encouragement; *e.g.*, *"you are a champion"*, *"we love you"*, *"you come first, we are sure you can do it"*.
- Gratitude: Present in recent models (*Ekman, 2004*; *Fredrickson, 2013*), the messages imply a sincere appreciation for sharing mental health-related content on social networks; *e.g.*, *"thank you for making this problem visible"*.
- Comprehension/empathy/identification: Present in *Plutchik (2001)*. They involve interest in and understanding of the message, including self-identification with the situation or context, putting oneself in the other person's shoes; *e.g.*, *"It happens to all of us and we've all been through times like this"*. They often provoke revelations of mental health problems by claiming to have gone through the same thing. *"I understand you so much… I want to get out of this depression"*.
- Sadness: This primary emotion (*Ekman, 2004*) is produced by events that are not pleasant and that denote heaviness. It includes many manifestations of pity for the person; *e.g.*, *"poor thing"* *"what a pity"*. There are also comments that include grief at the influencer's dismissal after her announcement of quitting social media; *e.g.*, *"You wil be missed"*.

- Anger/contempt/mockery: Initially, the classification of anger/angry and contempt/mockery following the primary emotions model (*Ekman, 2004*) was proposed. However, after a first classification, we decided to merge the categories, as anger was also present in cases of contempt/mockery. This category involves responses of irritation and attacks on the person as ridiculous and superficial; *e.g.*, *"uploading pictures of yourself crying is a higher level of ridiculousness"*; *"every time you cry you upload a picture… I'm so mad…"*; *"you're so disgusting"*; *"you're so disgusting"*; *"you're so ridiculous…"*; *"you're so ridiculous"*; *"you're so ridiculous"*.

## Dataset labelling methodology

The labelling procedure was divided into three phases: an initial phase with a pilot *corpus* (*N* = 787 comments), a subsequent phase with all comments on all posts (*N* = 21,151) and a final phase with the final *corpus* (*N* = 2,287) to be used by the machine learning algorithms, as it can be observed in Fig. S1.

Thus, the same methodology and steps were followed in the first two phases, as detailed below.

- *Data cleaning*: To ensure the relevance of the sample, this process removed comments that contained only acronyms, were made in other languages (not Spanish) or lacked linguistic coherence. In addition, comments naming other people have also been removed, except when the author of the post was tagged and included significant information.
- *Removal of emoticons*: To analyse the linguistic impact only.
- *Expert labelling*: The *corpus* was labelled by two experts blindly (psychologists, trained people) and a third expert reviewed the discrepancies. In case the third reviewer did not break the tie, these comments were discarded from the *corpus*. In case the third reviewer did not break the tie, these comments were discarded from the *corpus*. To carry out this process, a set of guidelines on labelling of the different categories was elaborated to train the experts. This guide included descriptions of the categories (emotions) explained in the previous section. In addition, certain labelling strategies were taken into account. For example, one strategy was to identify expressions such as "*Thank you*", which often indicated messages of gratitude, although this needed to be checked, as this was not always the case when accompanied by other expressions denoting different emotions. Similarly, expressions such as "*I love you very much*" often indicated messages of love. Thus, it was also observed that when someone said "*how sad*", it probably meant sadness. Finally, swear words or insults reflected comments of hatred or anger.

In the pilot *corpus*, a discrepancy of only 7.11% ($N = 56$) was obtained, indicating a strong inter-rate reliability among the experts who participated in the categorization process (*Hallgren, 2012*). The positive outcomes obtained from the pilot *corpus* serve as a robust foundation for reproducing the results on a more extensive sample, thereby guaranteeing the consistency of the conducted labelling process. Furthermore, these discrepancies predominantly stemmed from the categories of anger/angry and disgust/mockery. The decision was made to amalgamate both categories due to the frequent co-occurrence of anger in cases involving disgust/mockery/contempt. Some of the messages in which this happened were the following: "*All the same, it must be in fashion, haha…*" "*Taking pictures of yourselves crying to show others is like really unbalanced, seriously, doesn't the blood flow to your brain properly? Everyone goes through ups and downs, and there are genuine big problems and dramas in everyone's life*".

As for the second phase, the complete *corpus* ($N = 21,151$), the percentage of discrepancy was also around 9.19% (1,943). In this case, the greatest disparity was observed between comprehension/empathy/identification and love/admiration, because some messages included both emotions: "*You're strong, beautiful; never give up. I understand you perfectly, I went through the same thing too. May this crap of life never defeat us. I adore you*". Similar overlaps were observed with other emotions, like love/admiration and sadness, as exemplified in the following message: "*My heart breaks at the thought of you being sad. Lots of encouragement, those of us who love and admire you are waiting for you with open arms, much strength, and so much, so much love.*" Additionally, instances of overlap occurred between gratitude and comprehension/empathy/identification, as seen in the message: "*Thank you very much for sharing it; all of us going through this understand*

*you*". In cases where the third reviewer could not make a decision because there is more than one mixed emotion, these messages were discarded. There were also issues with messages that provided advice like "*you should disconnect from social media and rest,*" which were ultimately categorized as Neutral, as well as messages referencing a religious theme like "*may God be with you*".

In the third phase, corresponding to the final *corpus*, a meticulous curation process was implemented to form a representative *corpus* suitable for integration into our machine learning algorithms. Addressing the crucial aspect of class balance in data corpora, the reduction of the large *corpus* from 15,213 comments to 2,287 comments was executed through a systematic procedure. This reduction aimed to mitigate potential biases arising from imbalances in class distribution, as such imbalances can adversely affect the performance and predictive capacity of classification models. In fact, unbalanced classes cause a problem in predictive models as they tend to focus their attention on the majority class cases and do not correctly classify the remaining classes.

Thus, this process consisted of the next steps:

- *Removal of duplicate comments*: comments with repetitive content were removed, allowing a single message representative of each recurring topic to be retained. For example, messages containing phrases such as "*beautiful, we love you*" were condensed into a single message.
- *Distribution of comments across categories:* This distribution sought to achieve a more balanced representation of comments across all categories, to try to ensure that autistic learning models performed well for all classes. Although there are more comments from some classes than others, the distribution will be much more balanced across the whole *corpus*.
- *Random selection of comments:* comments were randomly selected from all posts based on the percentage distribution of each class in the final *corpus*.

### Dataset statistics

Figure S2 shows the descriptive statistics by percentage for each emotion represented in the final *corpus*. The most predominant emotion in the dataset is Comprehension/Empathy/Identification (28.8%), closely followed by Love/Admiration (28%), both positive emotions. In contrast, the minority emotions in the *corpus* are Gratitude (9.9%), Sadness (5.4%) and the Neutral class (7.6%). There is a certain imbalance between some categories in the *corpus*.

## MACHINE LEARNING CLASSIFICATION MODELS

### Random forest

Random Forest (RF) algorithms are machine learning models based on constructing multiple decision trees and combining their results to make more robust and accurate predictions. Each tree in the forest is trained independently with random subsets of data and features (*Breiman, 2001*). The advantages include its ability to handle large and

complex datasets, reduce the risk of overfitting, provide good generalization and be effective for classification problems (*Probst, Wright & Boulesteix, 2018*).

## A hybrid deep learning approach

A hybrid deep learning model is proposed, leveraging the potential of combining convolutional and recurrent layers. The convolutional layer performs tasks to streamline processing in the recurrent layer, thereby enhancing the overall efficiency of the model. Specifically, the proposed model consists of the combination of the following layers (Fig. S3):

- Embedding layer: It transforms Instagram comments, into vectors to be used by neural networks. In our case, we perform this process by means of word embedding, where each word is represented as a vector, applying the learning word embeddings technique using the *corpus* itself, to achieve more optimal results. Thus, the embedding layer can be viewed as a dictionary that maps integers, representing words, to dense vectors. When an embedding layer is initialised, these word vectors have random values, but during the training process, these vectors are gradually adjusted through the backpropagation technique. Once training is complete, the embedding space will reflect a structure adapted to the problem being addressed. In addition, to select the final vocabulary for word embedding, the problem *corpus* is used using the Information Gain (IG) method to preserve the terms with the highest occurrence. IG is preferable to absolute frequency as it measures the frequency of a word within a *corpus* category relative to its frequency in other categories. Absolute frequency, on the other hand, measures the overall frequency without distinguishing between classes, which does not provide relevant information (*Larose & Larose, 2014*; *Witten, Frank & Hall, 2011*).

- Convolution layer (one dimension): It applies filters to capture features and patterns on the input data. Its main function is to detect, learn and extract relevant patterns from the input data by convolution operations, thus eliminating part of the processing of the next layer, the recurrent layer (in our case LSTM). The hyparameters of this layer are the number of filters, and the filter size (kernel size) related to the length of the window and the activation function. Moreover, the rectified linear unit (ReLU) function will be employed.

- MaxPooling layer: This layer decreases the dimensionality of features from previous layers while preserving the most significant information. It operates on a data matrix to produce a smaller feature matrix that keeps the most significant features from the original matrix.

- LSTM layer: It is a recurrent neural network (RRN) that integrates long and short-term memory. The hyperparameters of this layer are number of neurons, dropout rate and recurrent dropout rate. The dropout technique disconnects neurons at each training stage until the optimal value is reached, so these parameters represent this fraction of neurons so that our network will learn just enough and avoid overfitting.

- Dense layer: It provides the model output. The hyperparameters of this layer are the number of neurons and the activation function. The number of neurons is the number of

classes to be predicted, in our case six neurons (five emotions, one neutral category). In addition, we select the Softmax activation function, widely used in multi-class classification problems.

As for optimisers and loss functions, the following have been selected:

- Categorical cross-entropy loss function for being largely adopted in multi-classification tasks.
- ADAM optimiser for its versatility, good performance and extensive use in similar problems (*Kingma & Ba, 2014*).

## Bidirectional Encoder Representations from Transformers (BERT) models

The Bidirectional Encoder Representations from Transformers (BERT) models are pre-trained deep learning models based on transformers architecture to learn bidirectional representations of text (*Devlin et al., 2018*). Using BERT for sentiment analysis involves pre-training the model on a large amount of data and then fitting it to a specific dataset. This allows models to acquire general knowledge of the language and then adjust to better understand the particularities of sentiment analysis in a specific domain, in our case mental health (*Wolf et al., 2019*). Then we have applied the next BERT classification models:

- RoBERTuito: A linguistic model for social networking texts in Spanish, based on RoBERTa and trained on 500 million tweets (*Pérez et al., 2021*). It outperforms other models for Spanish such as BETO, BERTin and RoBERTa-BNE (*ROBERTuito, 2022*).
- XML-RoBERTa (Danevi): A model created to improve multilingual comprehension models through unsupervised training with large amounts of multilingual text. They started from a trained model for the Masked Language Modeling (MLM) task and used a new *corpus* based on CommonCrawl (*Conneau et al., 2019*) data in 100 languages for pre-training. As a result, the new model achieved considerable improvements in multi-language classification tasks, up to 23% accuracy with respect to the original mBERT model. The model selected from the Hugging Face repository was trained for emotion classification in text by the user Danevi, so the model will be called by that name (*ROBERTa Emotion Twitter, 2021*).

# RESULTS AND ANALYSIS

## Data pre-processing and encoding

Data pre-processing is crucial to ensure the accuracy and efficiency of classification models. Our approach involves then next initial steps in text cleaning and pre-processing:

- Converting to lower case: to provide homogeneity in the text, so that transforming the same word to lower case, such as "Hello" and "hello", will cause them to be treated the same.

- To remove mentions (@), hashtags (#) and URLs: they are not relevant.
- To remove accents and diacritical vowels: to allow for greater homogenisation vocabulary and avoid including more noise in algorithms.
- To remove punctuation marks: they do not provide any meaningful information.
- To standardise slang/jargon: to transform these expressions into a standardised form of the language to improve data quality. For example, change "*d*" to "*de*" ("*from*" in English), "*xa*" to "*para*" ("*for*" in English), "*xq*" to "*por que*" ("because" in English).
- To delete Stopwords: empty words which have no meaning, such as articles, adverbs, prepositions, conjunctions and even some verbs. This elimination allows to focus on the most important information and improve classification performance, as they do not provide meaningful information.
- To apply stemming: a technique to reduce words to their base form. The aim is to simplify and standardise words to improve performance in text classification. For example, the Spanish words "*escribir*" ("*write*" in English) and "*escritor*" ("*writer*" in English) will be reduced to "*escri*". In this search, we will employ the NLTK's (*Snowballstemmer 2021*) in Spanish.

By contrast, BERT models include their own data preprocessing: character repetitions limited to a maximum of three, usernames and hastags are converted into a special token, and emojis are replaced by their textual representation. However, the RoBERTuito algorithm was tested with both data pre-processing, and the performance results were quite similar.

Another process consists of tokenization, process by which our texts (Instagram comments) are converted into inputs to be used by classification algorithms. The token can be thought of as the unit for semantic processing at the computational level. We use for RF and deep Learning models the TweetTokenizer tokenizer (*Bird, Klein & Loper, 2009*) while BERT models use their own tokenizer. Next, all the texts must be transformed into numbers (feature vectors), as machine learning models work with numbers and not with texts or words. In our case, we had to transform both the tokenised Instagram comments and the labels of our classes (emotions in our case).

To transform the messages into numbers, we employed the word embedding technique. In the case of deep learning, we applied the word embedding described in the previous section. In the case of BERT models, tokenisation generates hidden states that form word embeddings for each word, by multiplying these states by a learned weighting matrix. BERT's word embeddings are distinguished by their ability to capture context, causing the embedding of a word to fluctuate depending on its contextual usage within a sentence. In the case of labels, the One Hot Encoding technique has been used, which encodes the classes as a matrix, in which a "1" is placed in the column of the class (emotion) to which an Instagram comment belongs and a "0" in the other columns. In our case of five emotions (plus a neutral class), we will have a 6-column matrix. However, in the RF model, since it has only one output, instead of using a column for each variable and assigning

binary values (0 or 1), a single global variable (one output) is used, corresponding to a specific emotion.

The next step is to define two different datasets: the training set (70% data) and the validation set (30% data), these values being typical in other classification models (*Keras, 2023*). On the other hand, to find the best ensembles to train models we apply the k-fold cross-validation technique, which consists of performing k iterations and the model is trained and evaluated k times. Finally, we apply the EarlyStopping technique to regulate overfitting and prevent models from losing generalisation (*Keras, 2023*), so that the training of a model stops when the training loss metric stops improving. The classification models have been implemented in Python 3.1 using *Keras (2023)* and *TensorFlow (2023)*. libraries (*Abadi et al., 2016*). All the models were run on the Google Colab Platform, which allows running Jupyter notebooks on shared hardware provided by Google.

## Results and comparison of the classification models: RF, deep learning, BERT

This section describes the accuracy results of the classification models. In addition, we will consider other metrics to assess the performance of the models in each class separately, such as Recall, Precision and F1-score, which are especially important when the dataset is unbalanced. For this purpose, the best values of the hyperparameters of each model that optimise all the quality metrics will be sought.

### Results of the RF algorithm

In the RF model, the hyperparameters to optimise will be the number of trees and the hyperparameter *max_features*, which allows selecting the best features that optimise the splits in the decision trees. Therefore, the hyperparameters will be set in the following order: *max_features* and number of trees. Regarding the *max_features* parameter value, the results can be found in Table S2, where the choice of sqrt is highlighted, giving an accuracy of 46%.

Then, we proceed to find the best number of decision trees. Table S3 shows that 500 achieves an accuracy of 48% and there are no significant differences when changing this parameter.

In addition to global accuracy, Table S4 shows the results of precision, recall and F1-score for each class separately. It can be seen from the table that the precision for some classes is between 40% and 50%. The class with the highest level is Gratitude with a value of 87% and the worst predicted class is Sadness with poor results (40%).

### Results of the hybrid deep learning algorithm

To train and evaluate the performance of the model, different tests have been performed to optimise the accuracy metrics. These tests are:

- Adjustment of the number of filters and neurons in the convolutional and LSTM layers.
- Dropout rates and the learning rate for the Adam optimizer.
- Reduction of the *corpus* size.
- Adjustment of the batch size parameter.

As a first stage of the training phase, the number of neurons and filters in the layers will be varied to find the combination that gives the best results. The results are shown in Table S5, achieving the highest accuracy for 180 filters in the convolution layer and 96 neurons in the LSTM layer. A noteworthy feature is the sensitivity of the model to changes in both hyperparameters, which can be observed in the approximately 7% difference in accuracy between the highest and lowest values.

The second step is to vary the dropout rates (dropout rate and recurrent dropout) of the LSTM layer. According to Keras (2023), LSTM layers have two types of dropout rates, represented between 0 and 1. Table S6 shows that varying the values generates significant changes in the accuracy, and the best performance, 72.05%, is achieved with values 0.2 and 0.3, for the dropout and recurrent dropout rate parameters respectively.

The next step is to fix the learning rate parameter. The learning rate is quite important as it allows to control the speed of convergence as well as to avoid local minima. As a result of the study (Table S7), it can be observed that, depending on the value, there is a difference of approximately 7% in the results. The best value of the accuracy, 72.05% is achieved for a learning rate of 0.01.

The next training step is to reduce the total number of unique words in the *corpus* used in the model to increase computational efficiency. Nonetheless, there exists a trade-off between reducing the vocabulary and the potential loss of crucial information. Therefore, it is essential to conduct these reductions incrementally, for instance, in 100-word increments, as illustrated in Table S8. It becomes evident that using a smaller set of words, omitting those of lesser importance, enhances the model's accuracy. Nevertheless, as the word count continues to decline, accuracy eventually diminishes, since there will come a point when the vital words in the *corpus* begin to be discarded. Therefore, the size of the vocabulary that achieves the best accuracy, 72.5%, is 1,957.

Finally, the batch size defines the amount of data processed in each iteration of the model. The most common values have been tested, such as 32, 64, 128, 256 and 512, whose accuracy was 70%, 70.9%, 71.9%, 72.5% and 70.1%, respectively. While there are no significant differences, excessively high values can strain computational resources. Conversely, overly small values result in increased variability in the model evaluation metric. Thus, a batch size of 256 is chosen. Consequently, the optimized parameter configuration for the hybrid deep learning model is summarized as follows: Number of filters in the convolutional layer: 180, Number of neurons in the LSTM layer: 96, Dropout rate: 0.2, Recurrent dropout rate: 0.3, Learning rate: 0.01, Vocabulary size: 1,957, and Batch size: 256.

Although accuracy measures the percentage of total cases where the model has predicted correctly, in our case 72.5%, it is important to assess the model in each class separately. Thus, Table S9 shows the precision, recall and F1-score for each emotion. The results show that two of the positive polarities, obtain a better ranking (Love/Admiration and Gratitude); while the emotion Anger has such a high level of accuracy because it is the third most sampled emotion in the whole *corpus*. Furthermore, it can be observed that the results are quite good, with emotions such as Gratitude or Anger exceeding 80% and Love/Admiration with 77% precision. Moreover, the emotion of Comprehension/Empathy/

Identification, despite being the second most sampled, has an accuracy of 66%, probably because some messages may also contain the emotion Love/Admiration, which shows that, although the number of samples is decisive, it is not the only one that affects accuracy.

### Results of the BERT models: RoBERTuito and Daneni

Transformers provides the Trainer class to fine-tune the pretrained models on a specific dataset, and the optimal training parameters are the learning rate, the batch size and the epoch. Then, we tested the BERT models with the following parameter settings, that allows the best performance:

- Danevi (XML-RoBERTa): learning rate = 1,049 × 10−5, train batch size = 8, eval batch size = 2, num train epochs = 12
- RoBERTuito: learning rate = 7,557 × 10−5, train batch size = 16, eval batch size = 16, num train epochs = 6

Comparing both algorithms in Table S10, it can be stated that both models achieve very good results, although RoBERTuito is better than Danevi, obtaining an accuracy of almost 90% compared to 86% for Danevi. If we analyse the results for each class, we can see that the worst performing class is the neutral category (with very good values between 70–80%), while for all emotions the results are in both algorithms above 84%. In fact, Love/Admiration and Gratitude obtain the best metrics for both models, in contrast to the Sadness category, which is the emotion with the worst performance. Comparing both algorithms at each class, RoBERTuito predicts better than Danevi all emotions (including the Neutral class) except for Gratitude. In fact, RoBERTuito achieves high accuracies of 90% and above for all emotions except Sadness, although it also reaches very good results of 83%. This worse performance for the Sadness class may be due to the fact that it is the least minority class, with a much smaller number of samples than the other classes.

## DISCUSSION

Table S11 shows a comparison of the algorithms for the global accuracy metric. The best performance is obtained by the BERT models, RoBERTuito (90%) closely followed by Danevi (86%), and then the deep learning model with very good values of 72.5%. In contrast, the worst performance is obtained by RF, as expected. Overall, these results are quite satisfactory, as predicting emotions, especially in social media environments, is challenging due to:

- Subjectivity in the interpretation of emotions, since what one person considers to be joy, another person may interpret as a different emotion.
- Variability of emotions between individuals, as people express their emotions in different manners.
- Emotions can differ depending on the culture and or community to which a person belongs.

On the other hand, Table 12 shows the precision, recall and F1-score metrics for each class separately. Once again, the best results are obtained by the BERT models, followed by

the hybrid deep learning algorithm, and the worst performance corresponds to the RF model. Overall, RoBERTuito appears to perform well across most emotions, followed closely by Danevi model. In fact, for Love/Admiration, Sadness and Anger/Disrespect/Mockery, RoBERTuito outperforms other algorithms in precision, recall, and F1-score. For Gratitude, Danevi has the highest precision, while RoBERTuito has the highest recall and F1-score and for Comprehension/Empathy/Identification, RoBERTuito has the highest precision and F1-score, while Danevi has the highest recall.

Furthermore, the deep learning model shows the best performance when predicting Love/Admiration, Gratitude and Anger/Disrespect/Mockery. It is noteworthy that, despite the unbalanced distribution of the dataset, especially for Sadness and Gratitude emotions, RoBERTuito and Danevi predict the five emotions with high quality metrics, between 85–94% of precision, especially the former. Even for these emotions, the deep learning model performs quite well, between 60–85% for precision. This performance of all algorithms for precision and recall metrics is presented more visually in the radial plot in Figs. S4A and S4B.

Meanwhile, the confusion matrix makes it possible to visualise the effectiveness of the models and whether they confuse certain classes. Each column of the matrix represents the number of predictions in each class, while each row represents the instances in the actual class. Analysing the confusion matrix for the RoBERTuito and Danevi models (Figs. S5A, S5B), the classification efficiency for both models is very good, although the most confusion is produced between Love/Admiration and Comprehension/Empathy/Identification, where some comments belonging to Comprehension/Empathy/Identification are mistaken for Love/Admiration. Furthermore, the same performance can be observed for the deep learning model (Fig. S5C). This can happen because sometimes when showing love, empathy or comprehension can also be expressed, which can lead to confusion between the two emotions. Also, the confusion matrices indicate how models experience problems for the sadness emotion, especially the deep learning model. This may be due to the imbalance of classes in the dataset, where sadness is the minority.

In summary, in contexts where the detection of negative comments and emotions (sadness, hate, anger) as is the case of mental health in social networks, the consequences of false positives or false negatives have strong adverse implications, so achieving high levels of precision is of utmost importance. Therefore, the BERT models, especially RoBERTuito, exhibit very good results for this metric in all emotions (Fig. S4A), while showing strong performance in the recall and F1-score metrics (Fig. S4B). Even the deep learning model achieves very good results for most emotions, between 60–85% of precision.

Therefore, the modelling of these algorithms together with the mental health *corpus* allows the design of software tools based on machine learning algorithms to detect with high levels of accuracy the emotional response generated and thus be able to assess the impact and reach of mental health posts on social networks, never before addressed especially in social networks used by younger people. In fact, our proposal focuses, in a pioneering way with respect to other existing approaches, on analysing emotional response

in social networks, in contrast to existing research that focuses mainly on identifying types of psychopathologies.

Specifically, most of the studies focus on detecting symptoms of depression (*Liaw & Chua, 2022*; *Wongkoblap, Vadillo & Curcin, 2021*), suicidal behaviour (*Ramírez-Cifuentes et al., 2020*), or stigma (*Budenz et al., 2020*; *Jilka et al., 2022*; *Oscar et al., 2017*) towards different mental illness (bipolar disorder, schizophrenia, Alzheimer's), and most focus on Twitter, far removed from other more popular alternatives among young people such as Instagram. In addition, the Spanish *corpus* is the first *corpus* designed to analyse the impact of social responses to mental health content on social networks, especially on social networks for younger audiences, such as Instagram. These machine learning algorithms, alongside the mental health *corpus*, can be utilized by individuals, mental health professionals, or organizations to forecast the emotional responses to mental health-related social media posts. Consequently, they can formulate effective recommendations for addressing mental health in virtual environments and design more impactful awareness campaigns. This demonstrates the potential of this research and its results in a very worrying social context in our society.

## Limitations

In terms of potential limitations for the development of the study, it is worth highlighting the speed and temporality that are associated with social networks, since high-impact posts are often deleted or remain accessible for a short time, and social networks trends are particularly fast and unstable. To reduce this risk, a specific period has been selected, and posts of interest will be actively tracked throughout the study. Although showing a snapshot of a time is a limitation, the study is designed so that new posts can be integrated to feed back into the algorithms and improve or extend their effectiveness. In addition, another important challenge is the effectiveness of public participation strategies, especially in social networks, due to their inherent biases and the demographic diversity of users. Some groups may not actively participate in social networks, which could lead to biases in the collected data. This is an inherent risk of the research study, as it is primarily aimed at addressing mental health in virtual spaces.

Another potential limitation is the subjectivity of emotions and the lack of a universal classification, which may bias the prediction made by AI algorithms, but different classifications of emotions have been studied, using categorizations based on scientific findings and well-established in the literature that best fit our study. Another limiting issue is related to the balance between classes in labelled corpora, which can be prejudicial to the performance of classification algorithms. The presence of unbalanced classes poses a problem in predictive models as they tend to focus their attention on the majority class cases and do not correctly classify the rest of the classes. To address this problem, a careful selection process of the comments in the *corpus* has been carried out to try to mitigate this type of factor.

An additionally risk to be addressed is the bias of the *corpus* labellers, which is minimised as they have been trained in advance and the process was carried out by a set of two independent experts and reviewed by a third party (peer review process). The last

limitation of the research concerns the language of the comments, as only messages in Spanish responding to posts in this language were processed, so the context of the study is Spanish. Future analyses in other contexts and cultures can be analysed to compare cultural and language-dependent differences.

## CONCLUSIONS

This research is an innovative investigation focused on analyzing society's emotional response to mental health revelations made by celebrities on Instagram, nowadays considered one of the most popular platforms among young people. Another innovative is the use of artificial intelligence for detecting emotional responses to mental health issues in social networks, contrasting with existing research that has predominantly focused on the detection of psychopathological aspects. Although this analysis has been carried out in the Spanish context, its scope is global as social networks have a strong impact worldwide and our methodology is easily extrapolated to other social contexts.

First, a *corpus* has been designed based on Instagram posts made by Spanish celebrities talking about their mental health problems and manually labelled with emotions. This *corpus*, accessible online for researchers to use in their studies, is the first Spanish *corpus* prepare to analyze the impact of social responses to such issues. In addition, our research also addresses the modelling of machine learning algorithms to enable the efficient detection of emotions generated in this social context algorithms (*GitHub, 2024*). On the one hand, it should be noted that sentiment analysis in social media, faces significant challenges, such as the brevity of messages, lack of context or excessive use of informal language. These difficulties are even more accentuated regarding emotions due to different challenges, such as individual subjectivity in the interpretation of emotions, the diversity of emotions in different cultures or the way in which different people express and understand their emotions.

Despite these adverse conditions, the algorithms have demonstrated high accuracy in detecting emotions, since BERT models have achieved high accuracy metrics between 86–90% followed by the hybrid deep learning model with values around 72.5%. When analyzing the performance for each emotion, BERT models have shown good performance in all emotions, RoBERTuito with precision levels between 93% and 83.7% and Danevi between 94% and 81%. Both outperform the deep learning model, which records values between 85% and 55%. These findings are highly relevant, given the context of mental health in high-impact social media. In this scenario, the adverse implications of false positives and negatives are significant, especially in the detection of negative emotions (sadness, hate, anger). Consequently, these positive results drive the development of advanced computational tools based on machine learning to detect with high accuracy the emotional responses generated in real time. In this way, the aim is to effectively assess the impact of mental health on social networks.

In summary, the emotional analysis with machine learning of posts on mental health in social media, conducted by celebrities and influencers, is socially significant. Understanding these emotional responses can raise awareness and promote more meaningful content on social media, contributing to the promotion of ethically responsible

content about mental health. Artificial intelligence will enable the creation of effective campaigns to encourage a more responsible use of social media. In the future, we plan to expand the *corpus* with more Instagram posts to enhance balance and optimize algorithm performance further. Additionally, there is an intention to augment the *corpus* with content from other social networks, particularly TikTok. Given TikTok's increasing influence among young people, this expansion will broaden its potential applications.

### Funding
There was no funding for this study.

### Competing Interests
The authors declare that they have no competing interests.

### Author Contributions
- Noemi Merayo conceived and designed the experiments, performed the experiments, analyzed the data, performed the computation work, authored or reviewed drafts of the article, and approved the final draft.
- Alba Ayuso-Lanchares conceived and designed the experiments, performed the experiments, analyzed the data, prepared figures and/or tables, authored or reviewed drafts of the article, and approved the final draft.
- Clara González-Sanguino conceived and designed the experiments, performed the experiments, analyzed the data, prepared figures and/or tables, authored or reviewed drafts of the article, and approved the final draft.

### Human Ethics
The following information was supplied relating to ethical approvals (i.e., approving body and any reference numbers):

Regarding ethical considerations and data privacy, the study has been approved by the ethics and deontology committee of the University of Valladolid (PI 23-3365).

### Data Availability
Data and code are available at GitHub and Zenodo:

https://github.com/GCOdeveloper/Mental-Health-Dataset

NOEMI, M., Alba, A., & Clara, G.-S. (2024). Mental Health in Social Networks with Machine Learning Algorihtms [Data set]. Zenodo. https://doi.org/10.5281/zenodo.11202766

### Supplemental Information
Supplemental information for this article can be found online at http://dx.doi.org/10.7717/peerj-cs.2251#supplemental-information.

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
