# Peer review of "Machine learning and natural language processing to assess the emotional impact of influencers' mental health content on Instagram"

_PeerJ Computer Science, doi:10.7717/peerj-cs.2251_

## Round 0.1 · original submission · Minor Revisions

Dear Dr. Merayo,

Thank you for your submission to PeerJ Computer Science. Please read all reviews carefully and address all the queries.

Reviewer 1 ·

Basic reporting

Interesting work, I would like to thank the authors for this contribution. The methodology is clearly described, the results are presented with robust evidence, and the manuscript is generally well-written as well. However, I would like to offer some suggestions for improvement to the next version, please.

(1)
The study could benefit from a more comprehensive discussion of potential limitations in the methodology, and other factors that could affect the interpretation of results. This would provide readers with a more nuanced understanding of the study's findings and their implications.

(2)
Moreover, the study could delve deeper into the ethical considerations associated with the use of AI in that context, particularly focusing on algorithmic biases.

(3)
The introduction requires more depth, particularly in positioning the research within the context of Natural Language Processing (NLP). An improvement could involve integrating relevant studies that delve into the role of NLP in the mental health domain or healthcare in general. For instance:
https://doi.org/10.2196/15708
https://doi.org/10.5220/0010414508250832

(4)
More information about the data labelling process would have been helpful as it is a crucial aspect of this project. Sharing lessons learned in this regard with the research community could be beneficial. Therefore, a comprehensive explanation of the methodology used, including any challenges encountered and strategies employed, would be valuable, in my view.

(5)
Multiple references should be cited, please.
Devlin, J., Chang, M., Lee, K., & Toutanova, K. (2019). BERT: Pre-training of Deep Bidirectional Transformers for Language Understanding. In Proceedings of the Annual Conference of the North American Chapter of the Association for Computational Linguistics (NAACL-HLT).

Kingma, D.P. & Ba, J. (2015). Adam: a method for stochastic optimization. In Proceedings of the 3rd International Conference on Learning Representations (ICLR), San Diego, CA, USA.

Wolf, T., Debut, L., Sanh, V., Chaumond, J., Delangue, C., Moi, A., ... & Rush, A. M. (2019). Huggingface's transformers: State-of-the-art natural language processing. arXiv preprint arXiv:1910.03771.

Breiman, L. (2001). Random forests. Machine Learning, 45, 5-32.

Abadi, M., Barham, P., Chen, J., Chen, Z., Davis, A., Dean, J., ... & Zheng, X. (2016). {TensorFlow}: a system for {Large-Scale} machine learning. In Proceedings of 12th USENIX Symposium on Operating Systems Design and Implementation (OSDI 16) (pp. 265-283).

(6)
The writing is generally clear, but there is room for improvement in terms of paragraph length. Breaking down some of the longer paragraphs into smaller, more digestible parts would enhance readability and comprehension.

(7)
I suggest considering a title that effectively situates the study within the NLP context.

Experimental design

Well-designed study in general.

Validity of the findings

No concerns around the validity of the findings.

Additional comments

N/A

Reviewer 2 ·

Basic reporting

The paper discusses the rising concern surrounding mental health on social media, particularly focusing on the popularity of platforms like Instagram and the prevalence of mental health issues. It highlights the importance of understanding public attitudes towards mental health disclosures on social media, given the significant impact they can have. The paragraph also introduces the potential of Artificial Intelligence (AI) and machine learning in revolutionizing mental health analysis on social media platforms like Instagram, particularly through sentiment analysis techniques. The research aims to develop innovative methods for analyzing emotional responses to mental health disclosures on Instagram, contributing to a better understanding of public sentiment and potentially improving strategies for addressing mental health issues in online environments.

The study is imperative in the realm of exhaustive use of social media platforms. In general, the research provides a detailed overview of the intersection between mental health and social media, covering statistics, prevalence rates, and societal attitudes towards mental health disclosures.

In addition, the research employs a number of machine learning algorithms with extensive experiments to discuss the idea of exploring emotional impact. Furthermore, the study also suggests the recent state-of-the-art to explore the idea of impact generated by public figures to further strengthen the importance of this study.

Though this research has the potential to address mental health issues in online environments there are a number of limitations, including but not limited to,

The authors have provided an extensive experimental result, however, the study lacks in identifying the potential of this study in comparison to the state-of-the-art. This comparison will enhance the viability of this research.
The study does not include more recent state-of-the-art as I can see only a few studies from 2023 and 2024 are discussed.

Experimental design

No comment

Validity of the findings

No comment

---

## Round 0.2 · accepted · Accept

Thanks to the authors for your efforts to improve the work. This version satisfied the editor successfully.

Reviewer 1 ·

Basic reporting

Thanks for accommodating the feedback. I have no further comments.

Experimental design

I have no further comments.

Validity of the findings

I have no further comments.

Additional comments

I have no further comments.